# Effects of the Weather on the Seasonal Population Trend of *Aedes albopictus* (Diptera: Culicidae) in Northern Italy

**DOI:** 10.3390/insects14110879

**Published:** 2023-11-15

**Authors:** Marco Carrieri, Alessandro Albieri, Paola Angelini, Monica Soracase, Michele Dottori, Gabriele Antolini, Romeo Bellini

**Affiliations:** 1Centro Agricoltura Ambiente “G.Nicoli”, Sanitary Entomology & Zoology Department, 40014 Crevalcore, Italy; mcarrieri@caa.it (M.C.); rbellini@caa.it (R.B.); 2Regional Health Authority of Emilia-Romagna, 40127 Bologna, Italy; paola.angelini@regione.emilia-romagna.it (P.A.); monica.soracase@regione.emilia-romagna.it (M.S.); 3Istituto Zooprofilattico Sperimentale Della Lombardia e Dell’Emilia Romagna “B. Ubertini” (IZSLER), 25124 Brescia, Italy; michele.dottori@izsler.it; 4Environmental Protection Agency of Emilia-Romagna, Hydro-Meteo-Climate Service (ARPAE-SIMC), 40122 Bologna, Italy; gantolini@arpae.it

**Keywords:** *Aedes albopictus*, Bayesian model, MCMC, ovitraps, predictive models

## Abstract

**Simple Summary:**

In this research, we examined how temperature, rain, and relative humidity influence the seasonal population of *Aedes albopictus*, which was estimated using ovitrap data collected in the period 2010–2022 by the monitoring network of Emilia-Romagna region, Italy. The main results suggest that the winter–spring period (January to May) is crucial in determining the size of the first generation and overall seasonal growth of the species. We found that the monthly densities depend on various combinations of meteorological parameters.

**Abstract:**

Background: *Aedes albopictus*, the Asian tiger mosquito, has become a prevalent pest in Italy, causing severe nuisance and posing a threat of transmission of arboviruses introduced by infected travelers. In this study, we investigated the influence of weather parameters on the seasonal population density of *Aedes albopictus*. Methods: A Bayesian approach was employed to identify the best meteorological predictors of species trend, using the eggs collected monthly from 2010 to 2022 by the Emilia-Romagna regional monitoring network. Results: The findings show that the winter–spring period (January to May) plays a crucial role in the size of the first generation and seasonal development of the species. Conclusions: A temperate winter and a dry and cold March, followed by a rainy and hot spring and a rainy July, seem to favor the seasonal development of *Ae. albopictus*.

## 1. Introduction

*Aedes albopictus* (Skuse, 1894), commonly known as the Asian tiger mosquito, is a highly adaptable and invasive species with a global presence. Native to Southeast Asia, this mosquito has, over the past few decades, rapidly expanded its range far beyond its original habitat, posing significant public health concerns as both an invasive species and a vector of several infectious diseases.

The species has emerged as one of the most notorious invasive mosquito species in the world. Its global spread is attributed to various factors, including increased international trade and travel, as well as its remarkable ability to adapt to diverse environmental conditions. This mosquito species has successfully established populations in regions far from its native range, including parts of Europe, the Americas, Africa, and Oceania. In Europe, *Ae. albopictus* first appeared in Albania in the late 1970s, likely introduced via the international trade in used tires, which provide ideal breeding sites for these mosquitoes. Since then, it has rapidly expanded its distribution throughout southern Europe, including countries like Italy where it was introduced at the end of the 1980s, and is now the most abundant species in Italian urban areas [1] and a major threat to public health; indeed, it has been responsible for several arbovirus outbreaks in Europe: in France, an increasing number of autochthonous transmissions of dengue virus (DENV), chikungunya virus (CHIKV), and Zika virus (ZIKV) have been detected since 2010 [2], and in Italy, the species has caused two CHIKV epidemic events, in 2007 (Emilia-Romagna region) and 2017 (Lazio and Calabria regions) [3,4]. In Italy, the tiger mosquito causes a significant biting nuisance [5] and a constant risk of the spread of arboviruses introduced by infected travelers. For these reasons, it is the subject of intense control activities both in public and private areas [6]. Consequently, rigorous control measures, spanning both public and private domains, are underway [6], including the implementation of a National Arbovirus Surveillance program following the 2007 chikungunya outbreak, promoting a comprehensive, multidisciplinary approach to vector surveillance and management [7].

In this study, we conducted an analysis to discern the far-reaching impacts of climate, specifically temperature, rainfall, and humidity, on the seasonal density of *Ae. albopictus*. The interplay of temperature and precipitation exhibits both direct and indirect effects on population dynamics and the complete life cycle of *Ae. albopictus*, spanning from egg hatching to the emergence of adult mosquitoes [8,9,10]. The development of *Ae. albopictus* is notably influenced by temperature, with a range of 15–35 °C, an optimal growth zone at approximately 29–30 °C, and a critical lower threshold at 9–10 °C [11,12]. Elevated temperatures expedite the mosquito life cycle, resulting in reduced cycle duration and survival rates, whereas lower temperatures extend the life cycle and enhance survival rates. Temperature also significantly impacts female mosquitoes, affecting their flight activity, host-seeking behavior, and blood-feeding patterns [13,14]. These climate-driven variations in *Ae. albopictus* population dynamics and behavior have considerable implications for its invasive status worldwide, as it responds dynamically to climate fluctuations, potentially influencing its geographical spread and establishment in new regions.

Climate exerts various indirect effects on mosquito populations and control measures. While temperature directly impacts the larval cycle, relative humidity (RH) and rainfall play indirect but critical roles. Drought and evaporation in breeding sites decrease the carrying capacity for larval development [15,16,17]. Conversely, rain enhances the carrying capacity in urban areas but poses a threat during intense downpours by washing away young larvae, particularly in road drains, which are the primary breeding sites for *Ae. albopictus* [1]. Another indirect consequence of climate relates to the efficacy of mosquito control activities: elevated temperatures stimulate microbiological activity, hastening the degradation of larvicides, while lower temperatures reduce larval feeding activity, potentially diminishing larval insecticide consumption [18,19]. Moreover, intense rainfall not only affects insecticide persistence in road drains but may also wash away larvae, further complicating control efforts, based on personal observations. Relative humidity, serving as a bridging parameter between temperature and rain, influences the survival and field dispersion of adult mosquitoes [20]. These indirect climate effects collectively shape mosquito population dynamics and impact the effectiveness of control strategies, necessitating a comprehensive understanding for successful vector management.

The main purpose of this study is to provide an insight for interpreting predictive models concerning the seasonal trend of *Ae. albopictus*.

## 2. Materials and Methods

Study area. This study was performed in the Bologna (44°29′41″ N, 11°20′33″ E), Modena (44°38′49″ N, 10°55′30″ E), Cesena (44°8′22″ N, 12°14′46″ E), and Forlì (44°13′21″ N, 12°2′26″ E) municipalities, where the *Ae. albopictus* monitoring was conducted without interruptions from 2010 to 2022, according to the validation procedure included in the protocol of the Emilia-Romagna regional *Ae. albopictus* monitoring network (E-RMN_AA) [7,20] (Figure 1). In these four urban areas of the Emilia-Romagna region from 2010 to 2022, the control activities (monthly treatments of public drains) were similar in all areas and constant throughout the study period.

*Ae. albopictus* monitoring. In previous studies conducted in Northern Italy, the average *Ae. albopictus* egg density collected by ovitraps was correlated with the adult population estimated by human landing collection (HLC) [21]. In this study, data from the ovitrap collection from week 21 to 40 (summer period covered by large number of ovitraps sampled every 14 days and maximum development of the species) were used as a population estimate. GIS software (QGIS 3.22 Białowieża) was used to divide monitored urban areas into quadrants, and in each one, an expert entomologist has individuated, in a green-shaded area, the fixed monitoring stations following the regional guidelines [22]. In the field, trained technicians managed ovitraps, following the regional operating procedures. The ovitrap used (CAA14G model) was a 1.4 L cylindric black plastic container holding about 800 mL of *Bacillus thuringiensis var. israelensis* solution (1 mL of Vectobac 12AS/ovitrap) (Valent BioSciences, Sumitomo, Libertyville, IL, USA) and a strip of masonite (15 × 2.5 cm) as an egg deposition substrate. The monitoring network was organized with summer and winter ovitraps. Summer ovitraps were activated from week 21 to 40 and sampled every two weeks in Bologna (n. 110 ovitraps), Cesena (n. 75 ovitraps), Forlì (n. 60 ovitraps), and Modena (n. 60 ovitraps). The most representative stations in Bologna (20 ovitraps), Cesena (10 ovitraps), and Forlì (10 ovitraps) that reflected the average trend of urban areas were kept active year-round (winter ovitraps).

Meteorological data. Daily weather parameters (daily average air temperature at 2 m T, daily average relative air humidity at 2 m RH, daily cumulated precipitation R, and daily average global radiation RAD) were recorded throughout the course of this study in the four municipalities monitored (ERG5 datasets; Arpae—Regional Agency for Environmental Protection of Emilia-Romagna Region, [23]—https://dati.arpae.it/dataset/erg5-interpolazione-su-griglia-di-dati-meteo, accessed on 2 January 2023). In the three cold months (January, February, and March), we considered the number of frost days (F—the number of days with a minimum daily temperature < 0 °C).

Statistical analysis. The optimal number of ovitraps activated in each urban area was determined using the Taylor Power Law [22] at a precision level of D = 1.50–0.25. We used the 40 year-round active ovitraps to analyze the annual egg density trend and the 305 summer ovitraps to examine correlations between population trends and weather data. In multiple linear regression analyses, the inference of the dependent variable ignores the model uncertainty from the first selection of relevant predictors, resulting in overconfident parameter estimates that generalize poorly [24]. To overcome this limitation in the classical linear regression, we used a Bayesian multi-model linear regression, as described in Bergh et al. 2021 [24], a technique that retains all models for inference, weighting each model’s contribution by its posterior probability. The Akaike information criterion (AIC) weights were used to guide the choice of prior distribution on the regression parameters, and the Markov Chain Monte Carlo (MCMC) method was used to update these priors with the observed data in order to obtain the posterior distribution of the parameters. The MCMC algorithm involves simulating a Markov chain where each state corresponds to a set of parameter values. On each iteration, the algorithm draws a new set of parameter values from a proposal distribution and determines whether to accept or reject the proposal based on the ratio of the posterior probabilities of the new states to the posterior probabilities of the old states. In this study, 1000 samples were chosen for MCMC [25,26,27].

The lag influence of weather parameters on the population dynamics was mitigated using the monthly average number of eggs, and the number of eggs was standardized for each area with the average to reduce the effect of the differences in the carrying capacity between the areas, and we adopted the Spearman rank correlation, which is insensitive to transformation using monotonic functions. The Spearman correlation between the different weather parameters was calculated, and the standardized monitoring data were transformed by log(y + 1) to normalize the data and control the variance. The collinearity effect of variables was checked using VIF (Variance Inflation Factor) and Tolerance indexes. We used the open-source software R (version 4.2.2) and the BAS package [27,28] for the statistical analysis.

## 3. Results

Throughout the 13 years of monitoring across the four areas, a total of 43,259 oviposition substrates were collected from 305 fixed stations, resulting in a count of 1.22 × 10^7^ eggs. The average percentage of activated ovitraps was 82.10% (SD 19.68%) (Appendix A). A positive correlation was found between the monthly egg density data. The data collected in May, June, July, and August were correlated with those recorded in the subsequent month, and the correlation decreased during the summer. The correlation between monthly egg densities was strongest for June and July and weakest for August and September (Table 1).

The data collected showed a growing trend in the average number of eggs during the summer season (from the ending of May to the beginning of October) from 2010 to 2022 (R^2^ = 0.66, F_1,50_ = 38.55, *p* < 0.001) (Figure 2).

As typically happens in temperate regions, a bell-shaped trend of *Ae. albopictus* eggs, collected by monitoring during the summer season, was observed. Starting in early May, the seasonal pattern of eggs laid in the ovitraps (we have considered only the 40 ovitraps active throughout the year) was initially exponential, peaking in late July or early August, followed by a gradual decline (Figure 3).

In Table 2, we have listed the models that demonstrated the highest predictive adequacy. The first two columns represent P(M) and P(M|data), where P(M) denotes the prior probability of each model, and the column labeled P(M|data) signifies the posterior probability of each model following data observation, FM denotes the Bayes factor concerning the odds for each model, indicating the extent to which the odds in favor of a particular model increase after observing the data, and BF_10_ gives the relative predictive adequacy of the given model. The posterior coefficient provides information about each possible predictor in the linear regression model (Figure 4).

The number of frost days in winter (F), the average daily temperature and rain in March (T_3_ and R_3_), the temperature and RH in May (T_5_ and RH_5_), the rainfall in July (R_7_), and the mean temperature and RH in September (T_9_ and RH_9_) were the parameters that most influenced the summer average (weeks 21 to 40) of the *Ae. albopictus* population, where frost days (F), T_3_, R_3_, T_9_, and RH_9_ had a negative influence and T_5_, R_5_, and R_7_ had a positive effect (Equation (1) in Table 2 and Figure 4a). The winter parameters seem to have a strong influence on the seasonal trend of the *Ae. albopictus* population (Equations (2) and (3) in Table 2).

In Figure 4, we reported the posterior summaries of coefficients with a 95% credible interval between parameters considered for a better comprehension of their importance in the models. E_mean_ is the mean egg density expressed as n.egg/ovitrap/14dd; F indicates frost days; T indicates the average daily temperature; RH indicates the average relative humidity; R indicates the cumulated precipitation; and RAD indicates the daily average global radiation.

Furthermore, a good correlation was observed between the seasonal average and the number of frozen days in winter, the amount of rainfall, and egg density recorded in May (Equation (3) in Table 2). The influence of the winter period (frost days in January–March F, T_3_, and R_3_) on the egg density in May (E_May_) was evident but partially explains the variability of the data. We also identified significant positive interactions with May’s temperature (T_5_) and relative humidity (RH_5_) (Equation (4) in Table 2 and Figure 4b).

The egg density in June was positively correlated with the egg density in May (E_May_), temperature and rain in May (T_5_ and R_5_), and radiation in June (RAD_6_) and negatively correlated with the rain recorded in June (R_6_) (Equation (5) in Table 2 and Figure 4c). The densities of *Ae. albopictus* in May and June (E_MJ_), the rain (R_6_) and the radiation (RAD_6_) in June, and the temperature in July (T_7_) seemed to influence the mosquito population in July (Equation (6) in Table 2 and Figure 4d). The collection of *Ae. albopictus* eggs generally peaks in August (E_Aug_), and the egg densities in previous months (E_MJJ_ = May + June + July) play an important role. The July temperature (T_7_) seemed to have a negative impact on the August population (Equation (7) in Table 2 and Figure 4e). In September, when the females start to lay their winter eggs, the population of the previous months (E_MJJA_ = May + June + July + August) has a minor impact, while the temperature of August has a negative effect. The temperature recorded in September seemed to have a positive impact on the time of the development cycle and the flight capacity of gravid females (Equation (8) in Table 2 and Figure 4f).

Using winter and spring data, we tested Equation (2) (frost days, temperature, and rain recorded in March and temperature and RH in May), Equation (3) (frost days, rain, and egg density recorded in May) in Table 2 to predict the seasonal population density over the years, from 2010 to 2023, displayed in Figure 5.

The degree of accuracy is very good, especially considering the simplicity of the model. Furthermore, the expected correlation and prediction error (mean and absolute) for each year is still significant, with a Spearman’s correlation value of 0.82 (*p* < 0.001) and 0.87 (*p* < 0.001) for model 2 (Equation (2) in Table 2) and 3 (Equation (3) in Table 2), respectively.

The data for 2015 are an exception when compared to the two predictive models. In this case, the mosquito population trend may have been significantly influenced by the summer weather, especially the unusually high temperatures observed in July, which exceeded the period’s average by more than 2 °C and likely accelerated the development cycle (Equation (6) in Table 2).

## 4. Discussion

The growth of the *Ae. albopictus* population in spring (in temperate areas) or after the dry season (in tropical countries) is very rapid and archives varying densities depending on the carrying capacity of the environment, prevailing weather conditions, and control activities.

In this study, we analyzed, using the Bayesian approach, the effects of some weather parameters on the seasonal densities of *Ae. albopictus* in Italy (estimated by ovitraps of the monitoring network of the Emilia-Romagna region), where the population has a typically bell-shaped trend with initial exponential growth until it reaches a peak at the end of July–beginning of August and then declines in September/November, with the reduction in temperature and the deposition of a part of the eggs in embryonic diapause.

The winter–spring period (January to May) seems to play a relevant role in shaping the dimension of the first generation and the seasonal development of *Ae. albopictus* in Northern Italy. The number of frost days in winter, the amount of rainfall, and the temperature recorded in March seem to affect the number of *Ae. albopictus* eggs laid in May by the first-generation females developed from overwintering eggs. In this regard, we can make several hypotheses about how the weather conditions affect the *Ae. albopictus* population: (1) *Ae. albopictus* in temperate regions laid overwintering eggs in diapause (a form of dormancy linked to a strongly reduced metabolic activity) or in quiescence (a form reversible by flooding) [29]. In the Emilia-Romagna region, the two strains coexist (personal data), with the population overwintering in the form of eggs deposited in various breeding sites in a state of embryonic diapause or quiescence. The temperature sensitivity of these two types of eggs differs, and their survival rates during winter can be influenced by factors such as the exposure of the breeding sites and the duration of frost days [30]. (2) The heavy rainfall that can occur in March, coinciding with the hatching period of the overwintering eggs, has the potential to wash the young larvae from road drains, which are the main breeding sites of *Ae. albopictus* larvae in Italy [31,32]. (3) The low temperatures in March could delay the hatching of the overwintering eggs to a more favorable period, less susceptible to the abrupt temperature drops and heavy rainfall typical of this month. (4) The winter and early spring rain may keep the water levels in road drains high and prevent the activation of the egg-hatching process. The data collected in May, June, July, and August show a correlation suggesting that the population developed by overwintering eggs has a decisive impact on seasonal trends, while the influence of weather parameters, such as temperature and rainfall, on the population trend varies, with temperature having a positive effect in May and a negative effect in March and rainfall exhibiting a negative effect in March and a positive effect in July, particularly in September. In Italy, *Ae. albopictus* often breeds in water sources that are not solely dependent on rainfall, such as tanks, vases, or saucers filled for watering urban gardens or manholes within courtyards alimented by car washing. Moreover, rain can have an ambiguous influence on the population dynamics of *Ae. albopictus*: it may reduce the effectiveness and persistence of products utilized in larval control activities, trigger egg hatching, and flush away larvae by road drain basins. The total amount of rain, with its intensity and spatial distribution, may introduce complexities that are challenging to quantify and incorporate into a model. Humidity, which is influenced by factors such as rainfall, temperature, and wind, affects the survival and activity of adult mosquitoes. However, it does not appear to have a pronounced effect on the population dynamics of *Ae. albopictus* in Northern Italy.

The two simple models presented in this study (Figure 5) show a good accuracy in predicting the population size of *Ae. albopictus* in the Emilia-Romagna Region with weather and density data recorded in the winter and spring periods. However, it must be kept in mind that possible anomalous weather conditions which may happen in the summer can influence the trend of the mosquito population.

## 5. Conclusions

This study was based on robust monitoring data consolidated by regularly applying three levels of data quality validation [21], including a long time series of 13 years in areas where the mosquito control activities were similar and standardized. The data collected confirmed the impact of climate parameters on the *Ae. albopictus* population’s seasonal trend.

We speculate that winter temperature and spring weather conditions influence life cycle parameters of the first generation, which, in turn, can affect the seasonal population trends.

Our analysis and the obtained predictions are applicable to the Emilia-Romagna region, where our data were collected, and we cannot say to what extent population forecasting can be extended to other temperate regions. Nonetheless, our findings contribute to highlighting the relevance of the winter and spring weather parameters in the seasonal life cycle of *Ae. albopictus* in temperate regions.

Future studies on the effect of the control activities of the first-generation *Ae. albopictus* (the month of April in Italy) on the population trend need to be conducted in the field, with greater attention paid to the carrying capacity and density-dependent effects (additive, compensatory, and over-compensatory) on the mosquito population during the summer. Further studies can explore the impact of the greenhouse effect on winter and spring weather, evident in the last years in Italy, particularly in Po Valley, and the consequence on the mosquito density.

## Figures and Tables

**Figure 1 insects-14-00879-f001:**
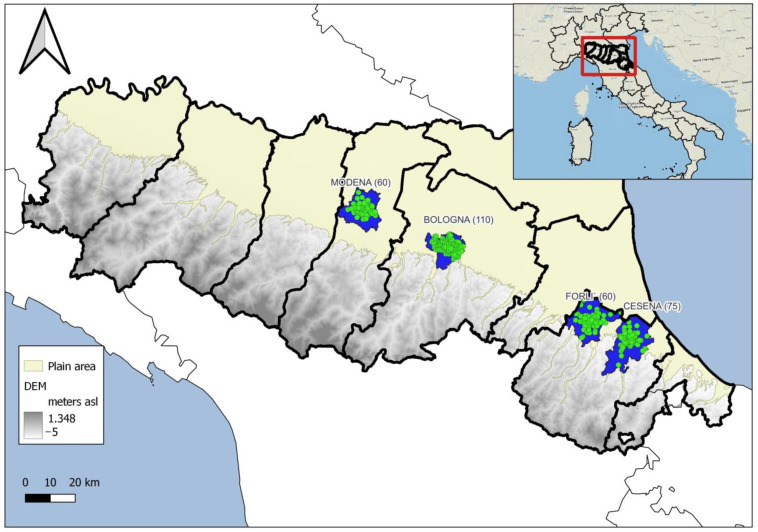
Study areas (blue polygons represent the monitored municipalities’ surface) and ovitraps (green points) in the Emilia-Romagna region, Italy. The number of ovitraps activated in each municipality is reported in brackets.

**Figure 2 insects-14-00879-f002:**
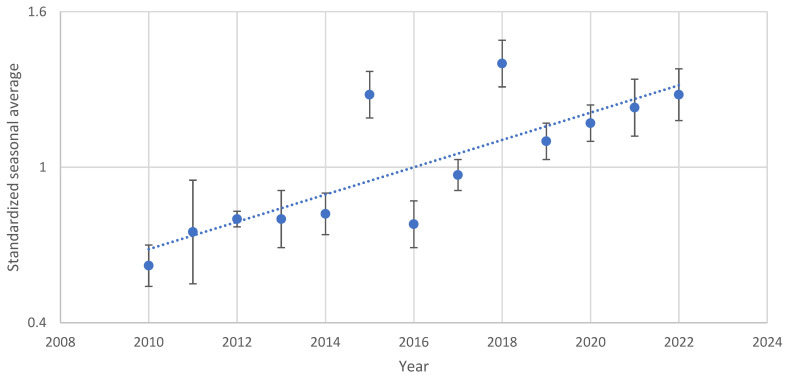
Standardized seasonal average of the number of eggs collected from 2010 to 2022 in the summer ovitraps (blue dots); bars represent the standard error and the dash line is the linear regression.

**Figure 3 insects-14-00879-f003:**
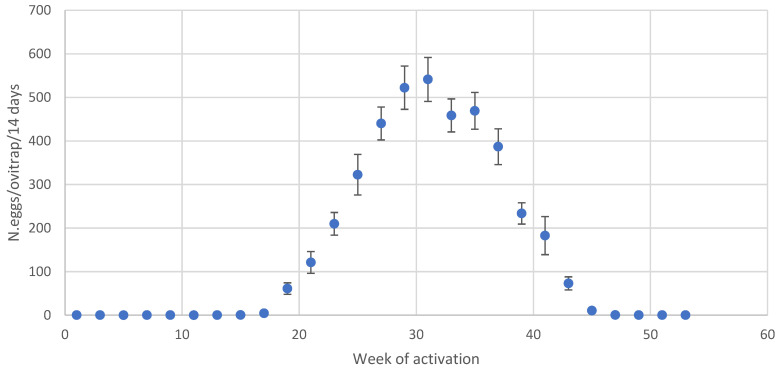
Seasonal trend of the number of eggs/ovitrap/14 days (blue dots) in Bologna, Cesena, and Forlì (40 ovitraps); bars represent the standard error.

**Figure 4 insects-14-00879-f004:**
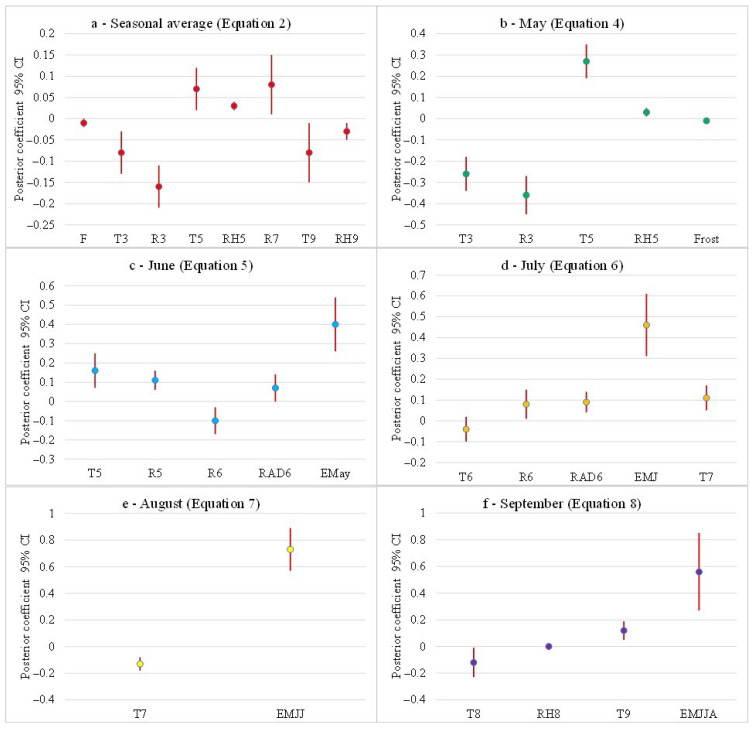
Posterior summaries of coefficients with 95% credible interval (CI) between weather parameters and the egg density; (**a**) seasonal average, (**b**) monthly average of May, (**c**) June, (**d**) July, (**e**) August, and (**f**) September. The equation number in each graphic’s title corresponds to the equation number in Table 2 F indicates the number of days with a minimum daily temperature < 0 °C between January and March; T, the average daily temperature; RH, the average relative humidity; R, the cumulated precipitation; RAD, the daily average global radiation.

**Figure 5 insects-14-00879-f005:**
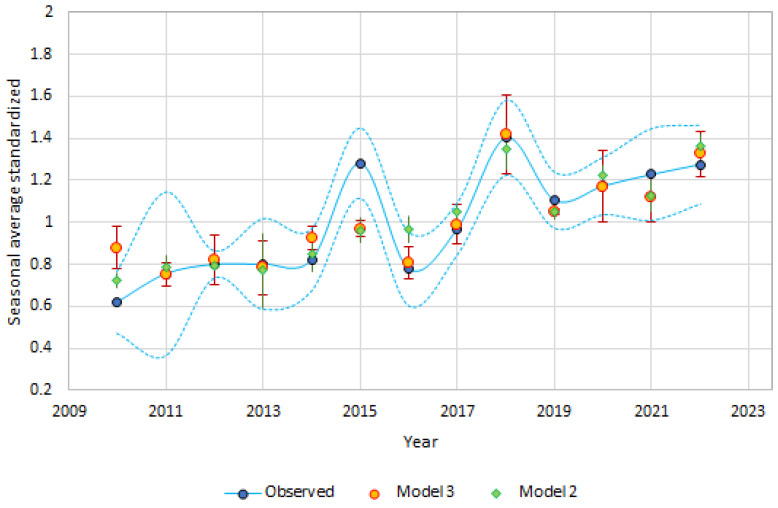
Observed data (dash lines represent the confidence limit 95%) and prediction of the average seasonal population density over the years, from 2010 to 2023, using Equations (2) and (3) (Table 2) and considering only the winter and spring data (bars represent ± confidence limit 95%).

**Table 1 insects-14-00879-t001:** The Spearman rank correlation between the monthly average of eggs collected in the four cities from 2010 to 2022.

	May	June	July	August
June	0.65 ***	—		
July	0.49 **	0.87 ***	—	
August	0.50 ***	0.63 ***	0.67 ***	—
September	0.18	0.31 *	0.31 *	0.56 ***

* *p* < 0.05; ** *p* < 0.01; *** *p* < 0.001.

**Table 2 insects-14-00879-t002:** Model comparison for seasonal (May to October) and monthly average densities from 2010 to 2022 (May to October) and weather parameters (Appendix A).

Equation of Models	*P* (*M*)	*P* (*M|data*)	*BF_M_*	*BF* _10_	*R* ^2^
(1) E_mean_ = 3.27 − 0.01F − 0.09T_3_ − 0.18R_3_ + 0.03RH_5_ + 0.08T_5_ + 0.09R_7_ − 0.09T_9_ − 0.04RH_9_	0.00	0.37	148.77	355,420.8	0.66
(2) E_mean_ = −0.80 − 0.01F − 0.08T_3_ − 0.15R_3_ + 0.08T_5_ + 0.02RH_5_	0.03	0.76	98.83	32,064.4	0.54
(3) E_mean_ = 0.58 − 0.006F + 0.07R_5_ + 0.37E_May_	0.20	0.84	20.93	96,959,746.54	0.56
(4) E_May_ = −2.15 − 0.01F − 0.27T_3_ − 0.37R_3_ + 0.28T_5_ + 0.03RH_5_	0.03	0.62	50.90	317,573,219,189.5	0.79
(5) E_June_ = −4.39 + 0.16T_5_ + 0.11R_5_ − 0.10R_6_ + 0.08RAD_6_ + 0.42E_May_	0.10	0.50	9.10	4,280,920,056.2	0.74
(6) E_July_ = −4.09 − 0.04T_6_ + 0.08R_6_ + 0.10RAD_6_ + 0.12T_7_ + 0.48E_MJ_	0.06	0.24	4.67	433,892,393.3	0.71
(7) E_Aug_ = 3.60 − 0.13T_7_ + 0.76E_MJJ_	0.09	0.42	7.30	2,095,791,453.6	0.66
(8) E_Sep_ = 1.39 − 0.14T_8_ − 0.00RH_8_ − 0.13T_9_ + 0.65E_MJJA_	0.04	0.23	7.08	390.42	0.40

## Data Availability

The data presented in this study are available on request from the corresponding author.

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
