# Peer review of "Effects of the Weather on the Seasonal Population Trend of Aedes albopictus (Diptera: Culicidae) in Northern Italy"

_insects, 2023, doi:10.3390/insects14110879_

Round 1
Reviewer 1 Report
Comments and Suggestions for Authors
The work by Carrieri et al. presents a Bayesian approach to the abundance (measured in egg counts) of Aedes albopictus in Northern Italy. The topic is of relevance due to the role of this mosquito as vector of diseases and the long-term and robustness of the dataset (4 towns, 13 years).
As I stated when I accepted the revision, although I am familiarized with multivariate analyses and GLMM I am not a specialist in Bayes approach, therefore in some passages I have more questions than suggestions. In general, the focus is on models for each month, and I feel that the very valuable long-term data (2010-2022) is not taken advantage of. For example, it would be really interesting to discuss which variables are associated with the rising trend depicted in Fig. 2. Also, an idea of why values in 2018 and 2018 are higher. I will therefore be willing to read an improved version of the MS considering this and the further suggestions below (I also attach a commented version of the MS with further input).
1. Climate change is a keyword, but is not mentioned once throughout the MS.
2. The introduction needs a little bit of work. Join sentences in more unified paragraphs, clearly state the purpose and influence of the work.
3. No need to include a point geolocalization for a town if you include a Figure. In Figure 1, please add the north row and geographic coordinates. The symbolism of the thick black lines, and the yellow, and black-gray shading is also missing. In the box, a little less zoom would be preferable (showing, for instance, the entire Italy in the regional context).
4 and more. Please see attached MS.

please thoroughly check the MS.
Author Response
Authors answers in red
Response to Reviewer 1 Comments
The work by Carrieri et al. presents a Bayesian approach to the abundance (measured in egg counts) of Aedes albopictus in Northern Italy. The topic is of relevance due to the role of this mosquito as vector of diseases and the long-term and robustness of the dataset (4 towns, 13 years).
As I stated when I accepted the revision, although I am familiarized with multivariate analyses and GLMM I am not a specialist in Bayes approach, therefore in some passages I have more questions than suggestions. In general, the focus is on models for each month, and I feel that the very valuable long-term data (2010-2022) is not taken advantage of. For example, it would be really interesting to discuss which variables are associated with the rising trend depicted in Fig. 2. Also, an idea of why values in 2018 and 2018 are higher. I will therefore be willing to read an improved version of the MS considering this and the further suggestions below (I also attach a commented version of the MS with further input).
- Climate change is a keyword, but is not mentioned once throughout the MS.
we change the key word with “climate”
- The introduction needs a little bit of work. Join sentences in more unified paragraphs, clearly state the purpose and influence of the work.
Done in the manuscript reviewed
- No need to include a point geolocalization for a town if you include a Figure. In Figure 1, please add the north row and geographic coordinates. The symbolism of the thick black lines, and the yellow, and black-gray shading is also missing. In the box, a little less zoom would be preferable (showing, for instance, the entire Italy in the regional context).
Map updated in the manuscript reviewed
4 and more. Please see attached MS.
Corrections done in the manuscript reviewed

Reviewer 2 Report
Comments and Suggestions for Authors
The article brings important information about the population dynamics of Aedes albopictus in Italy. However, the presented text is fragmented and the discussion superficial. Tables and figures are in excessive number, misconfigured and with incomplete titles. I would recommend:
1. Standardizing the size of the paragraphs, some of them are very short;
2. Line 80, replace Ae .albopictus by Aedes albopictus, as it is in the beggining of the phrase;
3. Improve the aesthetics of tables and figures.
4. Improve tables and figures tittles and include footnotes
5. The number of tables is excessive and some should be merged.
6. Table 2: Indicate the season;
7. Line 159: replace Ae .Albopictus by Ae. albopictus
8. Improve discussion, it is too short and should not be written as items, but in an essay form.
Author Response
Authors answers in red
Response to Reviewer 2 Comments
The article brings important information about the population dynamics of Aedes albopictus in Italy. However, the presented text is fragmented and the discussion superficial. Tables and figures are in excessive number, misconfigured and with incomplete titles. I would recommend:
- Standardizing the size of the paragraphs, some of them are very short;
Done in manuscript reviewed
- Line 80, replace Ae .albopictus by Aedes albopictus, as it is in the begining of the phrase;
Done in manuscript reviewed
- Improve the aesthetics of tables and figures.
Done in manuscript reviewed
- Improve tables and figures titles and include footnotes
Done in manuscript reviewed (map)
- The number of tables is excessive and some should be merged.
We have reduced the number of tables into one summary table
- Table 2: Indicate the season;
It is indicated (mean monthly and seasonal densities from may to october)
- Line 159: replace Ae .Albopictus by Ae. albopictus
Done in manuscript reviewed
- Improve discussion, it is too short and should not be written as items, but in an essay form.
Done in reviewed manuscript
Please see the attachment

Round 2
Reviewer 1 Report
Comments and Suggestions for Authors
Good work, the current version of the MS has been significantly improved. I particularly liked the new shape of the Discussion section.
Just a final suggestion, don´t lower the prize of your work! At the end of the Introduction you state: "The main purpose of the conducted study is to serve as a preliminary effort to provide valuable data for the development of predictive models concerning the seasonal trend of Ae. albopictus. "
Be more specific and give credit to your paper! You don´t provide data for the development of predictive models, you run and interpret the predictive models!
Comments on the Quality of English LanguageSmall grammar mistakes detected (e.g. the caption of Fig-1 should better read "...The number of ovitraps activated for each municipality is reported between brackets").
Author Response
Answers in red
Response to Reviewer 1 Comments (round2)
Good work, the current version of the MS has been significantly improved. I particularly liked the new shape of the Discussion section.
Just a final suggestion, don´t lower the prize of your work! At the end of the Introduction you state: "The main purpose of the conducted study is to serve as a preliminary effort to provide valuable data for the development of predictive models concerning the seasonal trend of Ae. albopictus. "
Be more specific and give credit to your paper! You don´t provide data for the development of predictive models, you run and interpret the predictive models!
Done in the reviewed manuscript
Small grammar mistakes detected (e.g. the caption of Fig-1 should better read "...The number of ovitraps activated for each municipality is reported between brackets").
Done in the reviewed manuscript
Last updates highlithed in yellow in the reviewed manuscript .

Reviewer 2 Report
Comments and Suggestions for Authors
The manuscript has improved, the number of figures and tables nas been reduced, and the discussion has been expanded as recommended. However, the manuscript still presents aesthetic concerns. There is a sequence very short paragraphs, the tables are non-standard and do not have a footer, preventing the compression of the abbreviations used, making it necessary to refer to the text to understand them. The figures also need editing for aesthetic improvement. I had already pointed out these issues in my first review, they were partially addressed, but the problem persists.
Comments on the Quality of English LanguageMinor revision needed.
Author Response
Answers in red
Response to Reviewer 2 Comments (round2)
The manuscript has improved, the number of figures and tables has been reduced, and the discussion has been expanded as recommended.
However, the manuscript still presents aesthetic concerns. There is a sequence very short paragraphs, the tables are non-standard and do not have a footer, preventing the compression of the abbreviations used, making it necessary to refer to the text to understand them.
Corrected in the reviewed manuscript
The figures also need editing for aesthetic improvement. I had already pointed out these issues in my first review, they were partially addressed, but the problem persists.
We have improved the aesthetic of figures and add more information on figures caption
Last updates highlithed in yellow in the reviewed manuscript .
